# HHD-ETHIOPIC
## A HISTORICAL HANDWRITTEN DATASET FOR ETHIOPIC OCR WITH BASELINE MODELS AND HUMAN-LEVEL PERFORMANCE

### ABSTRACT

This paper introduces HHD-Ethiopic, a new OCR dataset for historical handwritten Ethiopic script, characterized by a unique syllabic writing system, low-resource availability, and complex orthographic diacritics. The dataset consists of roughly 80,000 annotated text-line images from 1700 pages of $18^{th}$ to $20^{th}$ century documents, including a training set with text-line images from the $19^{th}$ to $20^{th}$ century and two test sets. One is distributed similarly to the training set with nearly 6,000 text-line images, and the other contains only images from the $18^{th}$ century manuscripts, with around 16,000 images. The former test set allows us to check baseline performance in the classical IID setting (Independently and Identically Distributed), while the latter addresses a more realistic setting in which the test set is drawn from a different distribution than the training set (Out-Of-Distribution or OOD). Multiple annotators labeled all text-line images for the HHD-Ethiopic dataset, and an expert supervisor double-checked them. We assessed human-level recognition performance and compared it with state-of-the-art (SOTA) OCR models using the Character Error Rate (CER) and Normalized Edit Distance (NED) metrics. Our results show that the model performed comparably to human-level recognition on the $18^{th}$ century test set and outperformed humans on the IID test set. However, the unique challenges posed by the Ethiopic script, such as detecting complex diacritics, still present difficulties for the models. Our baseline evaluation and HHD-Ethiopic dataset will encourage further research on Ethiopic script recognition. The dataset and source code can be accessed at https://github.com/ethopic/hhd-ethiopic-I.

## 1 INTRODUCTION

The gathering of historical knowledge heavily relies on analyzing digitized historical documents Lenc et al. (2021). In order to process a large number of these document images, automated tools that can convert images of the original handwritten documents into its digital format (e.g., with Unicode or ASCII texts ) are necessary Belay et al. (2020a); Wick et al. (2022). One such tool is Optical Character Recognition (OCR), which enables computers to extract textual information contained in images to then provide editing, translation, or search capabilities Cheng et al. (2022); Thuon et al. (2022). OCR systems often face difficulty in accurately recognizing historical documents, particularly those written in Ethiopic scripts, due to a shortage of suitable datasets for training machine learning models and the unique complexities of orthography Belay et al. (2019b); Niko-laidou et al. (2022). Typical historical handwritten Ethiopic manuscripts from different centuries are displayed in Figure 1.

The Ethiopic script, also known as the Abugida, Ge'ez, or Amharic script, is one of the oldest writing systems in the world, with a history dating back to the $4^{th}$ century AD Jeffrey (2004a). It is used to write several languages in Ethiopia and Eritrea, including Amharic, Tigrinya, and Ge'ez. The script has a unique syllabic writing system and is written from left to right. It contains about 317 graphemes, including 231 basic characters arranged in a 33 consonants by 7 vowels matrix, one special ($1 \times 7$) character, 50 labialized characters, 9 punctuation marks, and 20 numerals. The script's complexity is increased by the presence of diacritical marks, which are used to indicate vowel length, and other phonological features. Amha (2009); Meyer (2016); Meshesha & Jawahar

Figure 1: Sample historical handwritten document image from HHD-Ethiopic dataset: two-column $19^{th}$-century manuscript (left), one-column $20^{th}$-century manuscript (middle), two-column $18^{th}$-century manuscript (right).

Figure 2: The first two top rows of Fidel-Gebeta (the row-column matrix structure of Ethiopic characters): The first column shows the consonants, while the following columns (1-6) illustrate syllabic variations (obtained by adding diacritics or modifying parts of the consonant, circled in color). These modifications results a complex and distinct characters having similar shape, which making them challenging for machine learning models (see Appendix A)

(2007). For example, the first two consonant Ethiopic characters and their corresponding vowels formation are shown in Figure 2 ((refer Appendix A for an extended discussion).

The Ethiopian National Archive and Library Agency (ENALA) has collected numerous non-transcribed historical Ethiopic manuscripts from various sources, covering different periods starting from the $12^{th}$ century Wion (2006). These documents are manually cataloged and some are digitized and stored as scanned copies. They contain valuable information about Ethiopian cultural heritage and have been registered in UNESCO's Memory of the World program Belay et al. (2021a); Nosnitsin (2012). The manuscripts are mainly written in Ge'ez and Amharic languages, which share the same syllabic writing system.

To address the scarcity of suitable datasets for machine learning tasks in historical handwritten Ethiopic text-image recognition, we aim to prepare a new dataset that can advance research on the Ethiopic script and facilitate access to knowledge from these historical documents by various communities, including paleographers, historians, librarians, and researchers.

The main contributions of this paper are stated as follows.

- We introduce the first sizable dataset for historical handwritten Ethiopic text-image recognition, named HHD-Ethiopic.
- We assess the human-level performance of multiple participants in historical handwritten recognition to establish a baseline for comparison with machine learning models.
- We evaluate several state-of-the-art Transformer, attention, and Connectionist Temporal Classification (CTC)-based methods.
- We compare the prediction results of machine learning model with human-level performance in predicting the sequence of Ethiopic characters in text-line images.

The rest of the paper is organized as follows: Section 2 reviews the relevant methods and related works. Settings of human-level recognition performance and OCR models are described in section 3. Section 4 presents results obtained from the experiment and comparative analysis between the model and human-level recognition performance. Finally, in Section 5, we conclude and suggest directions for future works.

## 2 RELATED WORK

In this section, we briefly review related work in optical character recognition and highlight challenges we are facing in OCR of historical Ethiopic manuscripts.

### 2.1 OPTICAL CHARACTER RECOGNITION

Machine Learning techniques have been extensively applied to the problem of optical character recognition, see Breuel (2017); Wick et al. (2022); Zhong et al. (2015); Chen et al. (2021); Zhang et al. (2017) for a review. These achievements are possible due to the availability of numerous datasets designed for various document image analysis tasks across a variety of scripts:

Among these, we can mention IAM-HistDBFischer (2020), DIDA Kusetogullari et al. (2021), IMPACT Papadopoulos et al. (2013), GRPOLY-DB Gatos et al. (2015), DIVA-HisDB Simistira et al. (2016), ICDAR-2017 Dataset Sanchez et al. (2017), SCUT-CAB Cheng et al. (2022) and HJDataset Shen et al. (2020) as examples of historical and handwritten datasets. There are other datasets that can be used for printed and scene text-image recognition, including the ADOCR database Belay et al. (2019b), TANA Dikubab et al. (2022), OmniPrint datasets Sun et al. (2021), UHTelPCC Kummari & Bhagvati (2018), COCO dataset Veit et al. (2016), and TextCaps Sidorov et al. (2020), in addition to the historical and handwritten datasets mentioned previously.

Nowadays, segmentation-free OCR approaches Baek et al. (2019); Pal et al. (2007); Zhong et al. (2015) based on CTC Belay et al. (2020a); Du et al. (2022a); Breuel (2017); Messina & Louradour (2015); Wick et al. (2022); Shi et al. (2016a) attention mechanisms Lin et al. (2021); Qiao et al. (2020); Shi et al. (2018a); Zhang et al. (2017), and transformer-based models Barrere et al. (2021); Fang et al. (2021a); Mostafa et al. (2021) have become a popular choice among researchers and are widely used for text-image recognition (in both well-known and low-resourced scripts), as opposed to the traditional segmentation-based OCR approaches.

Researchers have reported remarkable recognition performance using these approaches for many well-known, such as Latin-based and Chinese, scripts ranging from historical to modern Barrere et al. (2021); Ly et al. (2019), and from handwritten to machine-printed Belay et al. (2021a).

However, there are other scripts that remain underresearched, particularly the Ethiopic scripts, which lack functional OCR systems. In the following sections, we briefly discuss the features of historical Ethiopic manuscripts and the challenges of developing OCR system for ancient Ethiopic manuscripts.

### 2.2 FEATURES OF HISTORICAL ETHIOPIC MANUSCRIPTS

There are various collections of ancient Ethiopic manuscripts in museums and libraries in Ethiopia and other countries. For example, the ENALA collection contains 859 manuscripts, the Institutes of Ethiopian Studies has 1500 manuscripts Nosnitsin (2012); A. Wion & Derat (2008), and the collections in Rome (Biblioteca Apostolica Vaticana), Paris (Bibliothèque nationale de France), and London (British Library) contain a total of 2700 manuscripts Nosnitsin (2012). These manuscripts were typically written on a material called Brana, which could vary in quality depending on the intended purpose or function of the book Mellors & Parsons (2002); Nosnitsin (2012).

Black and red were the most commonly used inks, with black reserved for the main text and red reserved for religious headings and names of significance. Figure 3 shows examples of historical Ethiopic manuscripts.

The manuscript layout can also vary and include formats such as three columns in the Synaxarion, one column for Psalms and prayer books, and two columns in liturgical books Bausi (2008); Nosnitsin (2012). The materials used for writing, including the pen and ink and the writing style, can also vary depending on the time period and region in which the manuscripts were produced. The use of punctuation marks is also very irregular (see Appendix A, Figure 11 for an extended discussion).

---

[1] https://expositions.nlr.ru/eng/ex_manus/efiopiia/efiopiia_letter.php

[2] https://upload.wikimedia.org/wikipedia/commons/2/2f/Sample_of_Ge%27ez_writing.jpg

[3] https://elalliance.files.wordpress.com/2013/11/world-history2.jpg

[4] https://www.w3.org/TR/elreq/images/kwk-mashafa-sawasew-page-268-typeface-change-for-emphasis.jpg

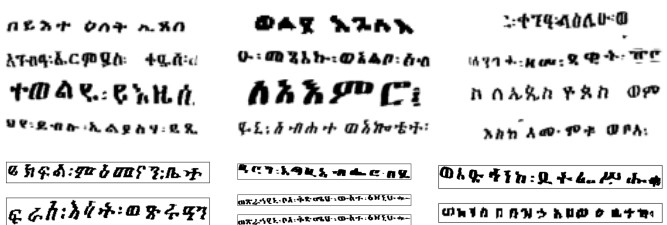

Figure 3: Examples of historical Ethiopic Manuscripts: (a) Two-column writing in liturgical books with decorated heading[1], (b) Two-column writing in liturgical books without decoration[2], (c) Three-column writing in the Synaxarion[3], (d) One column for Psalms and prayer books[4]. The Ethiopic script is written and read in the same direction as English, from left to right and top to bottom.

Historical documents, such as Ethiopic manuscripts, often have artifacts like color bleed-through, paper degradation, and stains, making them more challenging to work with than contemporary, well-printed documents Esposito (2013). Some major challenges in recognizing historical Ethiopic manuscripts include: (i) the complexity of character sets and writing system, which consists of over 317 distinct but similar-looking indigenous characters (see Figure 2 and details are given in Appendix A); (ii) variations in writing styles, including handwriting and punctuation, which can vary greatly among individuals and over time, affecting model accuracy; and (iii) a shortage of labeled data for training machine learning algorithms for Ethiopic script recognition.

Therefore, in this paper, we aim to tackle the challenges in recognizing the Ethiopic script by creating a new dataset called HHD-Ethiopic which is composed of manuscripts dating from the $18^{th}$ to $20^{th}$ centuries. We also evaluate various state-of-the-art OCR models and compare their performance against human-level benchmarks.

## 3 DATASET AND BASELINE METHODS

In this section, we provide an overview of our work, focusing on two key aspects: the detailed characteristics of our new dataset (subsection 3.1) and the benchmark methods employed. Our dataset, comprehensively outlined, includes essential details such as size, composition, data collection, and annotation process. It serves as a valuable resource for evaluating historical handwritten Ethiopic OCR. Additionally, we present the benchmark methods, including human-level recognition performance and baseline OCR models (subsection 3.2).

Table 1: Summary of the training and test text-line images

| Type-of-data | Pub-date-of-manuscript | #text-line images | remark |
|---|---|---|---|
| Training-set | 90% of (A+B+C) | 57,374 | real |
| Test-set-I (IID) | 10% of (A+B+C) | 6,375 | real |
| Test-set-II (OOD) | 100% of (D) | 15,935 | real |

A= Unknown pub. date, B= $20^{th}$ century, C= $19^{th}$ century, D= $18^{th}$ century manuscript

Figure 4: Sample historical handwritten Ethiopic text-line images from HHD-Ethiopic

## 3.1 HHD-ETHIOPIC DATASET

The HHD-Ethiopic dataset consists of 79,684 text-line images with their corresponding ground-truth texts that are extracted from 1,746 pages of Ethiopic manuscripts dating from $18^{th}$ to $20^{th}$ centuries. The dataset includes 306 unique characters (including one blank token), with the shortest text comprising two characters and the longest containing 46 characters. These 306 characters are not distributed equally; some occur more frequently due to the nature of the script, being widely used in the writing system. For example characters such as መ, ነ, ስ, በ, ት, ይ, ም, ላ, ር, ተ, ሞ, ብ, ከ, ል, etc are among the most frequent characters, whereas characters like ፐ, ፓ, ዣ, ጛ, ኺ, ጿ, ሿ, ፚ, ዿ, ሿ, etc are notably infrequent, occurring almost below a count of 10. In response to this issue of underrepresentation, we have generate a separate synthetic text-line images from these characters (see the Appendix section B.1 for an extended discussion)

The training set includes text-line images from recent manuscripts, primarily from the $19^{th}$ and $20^{th}$ centuries. We created two test set: the first one consists of 6375 images that are randomly selected using a sklearn train/test split protocols[5], from a distribution similar to the training set, specifically from $19^{th}$ and $20^{th}$ century books. The second one, with 15,935 images, is drawn from a different distribution and made of $18^{th}$ century manuscripts (see Table 1 for the splitting processes and size of the each set). The goal of the first test set is to evaluate the baseline performance in the IID (Independently and Identically Distributed) setting, while the second test aims to assess the model's performance in a more realistic scenario, where the test set is OOD (Out-Of-Distribution) and different from the training set.

To perform preprocessing and layout analysis tasks, such as text-line segmentation, we utilized the OCRopus[6] framework. For text-line image annotation, we developed a simple tool with a graphical user interface, which displays an image of a text-line and provides a text box for typing and editing the corresponding ground-truth text. Additionally, we employed this tool to collect predicted text during the evaluation of human-level performance.

A team of 14 people participated in creating the HHD-Ethiopic datasets, with 12 individuals tasked with labeling and the remaining two individuals responsible for reviewing and ensuring the accuracy of the alignment between the ground-truth text and text-line images, making any necessary corrections as needed. To ensure the accuracy of the annotations, participants were provided with access to reference materials for the text-lines, and all of them were familiar with the characters in the Ethiopic script. Table 1 and Figure 4 provide a summary of the dataset and show sample text-line images of the HHD-Ethiopic dataset, respectively (see Appendix B for an extended discussion).

## 3.2 SETTINGS FOR HUMAN-LEVEL PERFORMANCE AND BASELINE MODELS

To establish a baseline for evaluating the performance of models on the HHD-Ethiopic OCR dataset, we propose two approaches: (i) **Human-level performance** and (ii) **Sequence-to-sequence models**.

The human-level performance serves as a benchmark for evaluating and comparing the recognition performance of machine learning models on historical handwritten Ethiopic scripts and provides insights for error analysis. To calculate the human-level recognition performance, 13 independent annotators were hired and divided into two groups. It is important to note that these individuals are different from those mentioned in section 3.1. The first group transcribed text-line images from the first test set, which consisted of 6375 randomly selected images from the training set. The second group transcribed the second test set of 15935 images from the $18^{th}$ century. Each text-line image was predicted by multiple people (i.e nine for Test-set-I and four for Test-set-II). The annotators were already familiar with the Ethiopic script, and they were explicitly instructed to carry out the task without using any references. The predicted texts by each annotator, along with comprehensive details of the data collection and annotation process, is documented as metadata for future reference. The second reference point involves various state-of-the-art OCR models, which includes CTC, attention and transformer-based methods. The CTC-based models employ a combination of Convolutional Neural Networks (CNN) and Bidirectional Long Short-Term Memory (Bi-LSTM) as an encoder and CTC as a decoder, and is trained end-to-end with and without an Attention mechanism (see Appendix C for an extended discussion). In addition, for the attention-

---

[5]https://scikit-learn.org/stable/modules/generated/sklearn.model_selection.train_test_split.html
[6]https://github.com/ocropus/ocropy

based baseline, we utilize ASTER Shi et al. (2018a), and for the transformer-based baselines, we adopt ABINet as proposed by Fang et al. (2021a).

Moreover, we use Bayesian optimization (see e.g., Balaprakash et al. (2018a); Egele et al. (2022) for a review) to optimize the hyperparameters of the CTC-based models. Optimizing hyperparameters involves finding an optimal setting for the model hyperparameters that could result in the best generalization performance, without using test data. Considering the trade-offs between model performance and computational cost, we use a small subset of the training set to optimize the hyperparameters of models (see, e.g.,Bottou (2012) for a review), and then train the model on the full training set using the optimal hyperparameter settings.

We used the Character Error Rate (CER) Belay et al. (2020a); Graves et al. (2006a) and Normalized Edit Distance (NED)Chng et al. (2019) as our evaluation metric for both the OCR models and human-level recognition performances (see appendix C, equation 3 and 4 for an extended discussion).

## 4 EXPERIMENTAL RESULTS

Our objective is to perform a fair comparison between human and machine performance on historical handwritten Ethiopic scripts recognition task. This comparison is intended to showcase the utility and value of our new HHD-Ethiopic dataset, evaluate human recognition capabilities, and highlight any advancements made by baseline OCR methods.

### 4.1 HUMAN-LEVEL PERFORMANCE

As previously discussed in Section 3.1, the ground-truth text was annotated by multiple people and double-checked by supervisors who were familiar with Ethiopic scripts. For this phase, new annotators who were also familiar with Ethiopic characters were selected and instructed not to use any reference materials. The reviewer of both the training and test sets was permitted to use reference materials. However, in contrast to the training set, the test sets were reviewed by an expert in historical Ethiopic documents.

Table 2: The human-level recognition performance in Character Error Rates (CER) and Normalized Edit Distance(NED)

| Type-of-test data | Year-of-Pub | Annotator-ID | CER | NED |
|---|---|---|---|---|
| IID | $19^{th}$ & $20^{th}$ | Annot-I | 29.02 | 27.67 |
| | | Annot-II | 27.87 | 25.89 |
| | | Annot-III | 29.93 | 28.16 |
| | | Annot-IV | 29.16 | 27.80 |
| | | Annot-V | 26.56 | 24.56 |
| OOD | $18^{th}$ | Annot-VI | **25.39** | **23.78** |
| | | Annot-VII | 29.26 | 28.08 |
| | | Annot-VIII | 25.95 | 24.78 |
| | | Annot-IX | 51.03 | 25.46 |
| | | Annot-X | **33.20** | **30.77** |
| | | Annot-XI | 54.33 | 52.20 |
| | | Annot-XIII | 39.96 | 35.90 |
| | | Annot-XIV | 45.06 | 39.89 |

To measure the human-level recognition performance, multiple annotators were asked to predict the text in the images and then their character recognition rates were recorded. The best annotator on Test-set-I scored a CER of 25.39% and an NED of 23.78% on Test-set-I, and a CER of 33.20% and an NED of 30.73% on Test-set-II. In contrast, the average human-level recognition performance was a CER of 30.46% and an NED of 26.32% on Test-set-I, and a CER of 35.63% and an NED of 38.59% on Test-set-II. We used the best human-level recognition performance as a baseline for comparison with SOTA machine learning models' performance throughout this paper. Table 2,

shows the human-level recognition performance on both test sets, based on assessments from nine annotators on Test-set-I and four on Test-set-II.

## 4.2 BASELINE OCR MODELS

This section presents the results obtained from the experimental setups detailed in Section 3. Firstly, we present the results of the CTC-based OCR models previously proposed for Amharic script recognition Belay et al. (2020a; 2021a), followed by the results of other state-of-the-art models Du et al. (2022a); Fang et al. (2021a); Shi et al. (2016a; 2018a) validated in Latin and/or Chinese scripts.

The experiments conducted using the CTC-based models previously proposed for Amharic script were categorized into four groups:

- **HPopt-Plain-CTC**: plain-CTC (optimized hyper-parameters)
- **Plain-CTC**: Plain-CTC
- **HPopt-Attn-CTC**: Attention-CTC (optimized hyper-parameters)
- **Attn-CTC**: Attention-CTC

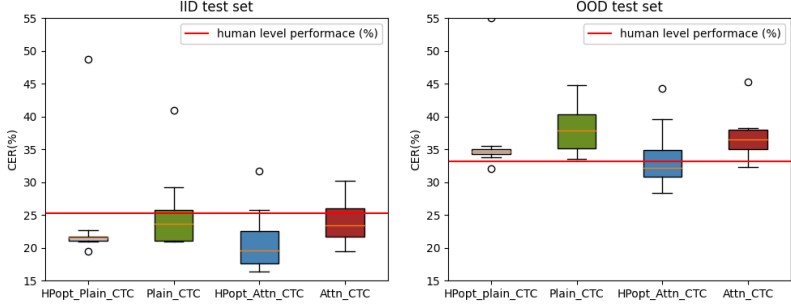

Figure 5: Box Plot comparison of variance in the recognition performance of CTC-based models and human level performance from ten experiments with varying random weight initialization and training sample orders on Test-set-I (IID) (left) and Test-set-II(OOD) (right). The results demonstrate that HPopt-attn-CTC outperforms all other CTC-based methods and surpassing human-level recognition on both test sets.Since the second group of models is too complex, we conducted individual experiments. Therefore, instead of a Box Plot, a learning curve is presented in Figure 6.

In all the CTC-based setups, to minimize computational costs during training, we resized all the text-line images to 48 by 368 pixels. We used 10% of the text-line images randomly drawn from the training set for validation. As previously discussed, in Section 3, we have two test sets: (i) Test-set-I, which includes 6375 text-line images randomly selected from $19^{th}$, $20^{th}$ century manuscripts and other manuscripts with unknown publication dates, and (ii) Test-set-II, a text-line images that are drawn from a different distribution other than the training set, which includes 15935 text-line images from $18^{th}$ century Ethiopic manuscripts only. The HPopt-Attn-CTC baseline model achieved the best CER of 16.41% and 28.65% on Test-set-I and Test-set-II, respectively (see Table 3 for details).

The results depicted in Figure 5 demonstrate that the CTC-based OCR models outperform human-level performance on Test-set-I in all configurations. However, only the HPopt-Attn-CTC model can surpass human-level performance, while the other configurations achieve comparable or worse results compared to human recognition on Test-set-II. Test-set-I was randomly selected from the training set, while Test-set-II consisted of $18^{th}$ century manuscripts and represented out-of-distribution data. This disparity in performance is to be expected, as machine learning models typically perform better on samples that are independently and identically distributed rather than those in an out-of-distribution setting. The repeat experiments aimed to capture the variability in the performance of the models due to random weight initialization and sample order.

HPopt-plain-CTC exhibits consistent variability across the 10 experiments due to the benefits of hyper-parameter optimization and a simplified architecture without attention mechanisms. The systematic fine-tuning of hyper-parameters, coupled with a simpler model architecture, resulted in stable and predictable performance throughout the experiments. In contrast, HPopt-attn-CTC achieved

the lowest error despite some variability in certain trials, demonstrating its robustness across ten trials (see Table 3). The optimized hyperparameter configuration significantly improved recognition accuracy compared to non-optimized settings on both test sets, highlighting the importance of hyperparameter tuning for superior performance beyond relying solely on prior knowledge or trial-and-error approaches.

The second category of baseline OCR models assessed using our HHD-Ethiopic dataset comprises state-of-the-art models, including CRNN Shi et al. (2016a), ASTER Shi et al. (2018a), ABINet Fang et al. (2021a), and SVTR Du et al. (2022a). Considering our available computing resources, all these models were trained for 25 epochs. The learning curve illustrating the recognition performance of these models on the IID and OOD test sets is depicted in Figure 6. In this group, the SVTR and

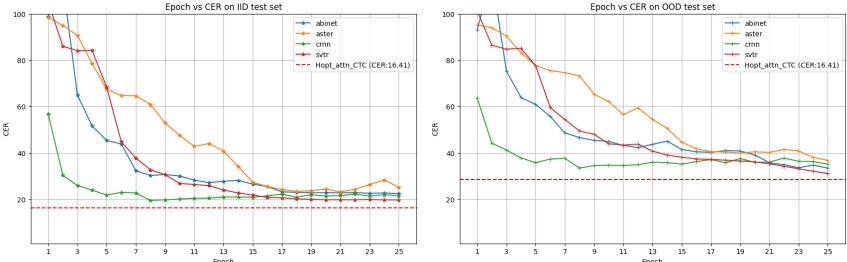

Figure 6: Learning curve on IID and OOD test data. CER on IID test set (left), CER on OOD test set (right) across 25 epochs for ASTER, ABInet, SVTR, and CRNN models. In all plots, the red horizontal line represents the CER value of the Hopt-attn-CTC network on IID and OOD data respectively.

ABINet models achieved the highest performance, with both models showing nearly equivalent results within a 1% difference during evaluation. As shown in Table 3, compared to the CTC-based models, the attention and transformer-based models exhibit larger number of parameter (see Appendix C for an extended discussion).

Table 3: A summary of baseline models and their recognition performance on Test-set-I (IID, 6k) and Test-set-II (OOD, 16k) using CER and NED. The table includes model parameters measured in millions (M).

| Methods | #Model-Parms | Type-of-test data | CER | NED |
|---|---|---|---|---|
| Plain-CTCBelay et al. (2020a) | 2.5M | IID | 20.88 | 19.09 |
| | | OOD | 33.56 | 31.9 |
| Attn-CTC Belay et al. (2021a) | 1.9M | IID | 19.42 | 21.01 |
| | | OOD | 33.07 | 32.92 |
| HPopt-Plain-CTC | 4.5M | IID | 19.42 | 17.77 |
| | | OOD | 32.01 | 29.02 |
| HPopt-Attn-CTC | 2.2M | IID | **16.41** | **16.06** |
| | | OOD | **28.65** | **27.37** |
| CRNN Shi et al. (2016a) | 8.3M | IID | 21.04 | 21.01 |
| | | OOD | 29.86 | 29.29 |
| ASTER Shi et al. (2018a) | 27M | IID | 24.43 | 20.88 |
| | | OOD | 35.13 | 30.75 |
| SVTR Du et al. (2022a) | 6M | IID | 19.78 | 17.98 |
| | | OOD | 30.82 | 28.00 |
| ABINet Fang et al. (2021a) | 23M | IID | 21.49 | 18.11 |
| | | OOD | 32.76 | 28.84 |
| Human-performance | | IID | 25.39 | 23.78 |
| | | OOD | 33.20 | 30.77 |

[7]https://matplotlib.org/stable/api/_as_gen/matplotlib.pyplot.boxplot.html

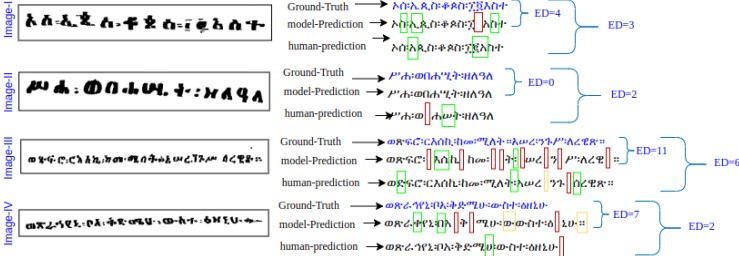

Figure 7: Sample human-machine recognition errors per text-line image from the Test-set-I. Deleted characters are marked in red, while substituted and inserted characters are marked by green and yellow boxes, respectively. The inner ED denotes the Edit distance between the ground-truth and model prediction, while the outer ED denotes ground-truth to human prediction Edit distance.

Based on Figure 7 and our experimental observations, we observed distinct error patterns between humans and models: both exhibit substitution errors, but the model tends to make a higher number of insertions and deletions. This highlights the imperfection of the baseline OCR models in terms of sequence alignments. Furthermore, our study found that the evaluated baseline OCR models were highly effective, surpassing human-level recognition performance on Test-set-I. However, only a few models achieved better recognition performance on Test-set-II. Compared to other methods, the HPopt-Attn-CTC model has achieved the best recognition accuracy on both datasets.

The baseline models evaluated in this study comprise CTC-based models previously proposed for the Amharic script, alongside five state-of-the-art attention and transformer-based models validated using English and Chinese scripts. These models could serve as references for evaluating the effectiveness of advanced models in recognizing historical handwritten Ethiopic scripts. Each of the CTC-based models previously proposed for Amharic script underwent ten experiments. In contrast, the other models, although trained for only single experiments and fewer epochs, achieved comparable outcomes. In addition, among the CTC-based models, the optimized hyperparameters model demonstrates superior performance, benefiting from fine-tuning and reduced overfitting. The reported results and dataset serve as a benchmark for future research in machine learning, historical document image analysis, and recognition, while the analysis of human-level recognition performance enhances our understanding of the dataset.

## 5 CONCLUSION

In this paper, we presented a novel dataset for text-image recognition research in the field machine learning and historical handwritten Ethiopic scripts. The dataset comprises 79,684 text-line images obtained from manuscripts ranging from the $18^{th}$ to $20^{th}$ centuries and includes two test sets for evaluating OCR systems in both the IID (Independent and Identically Distributed) and OOD (Out-of-Distribution) settings. We provided human-level performance and baseline results using CTC, attention and transformer based models to aid in the evaluation of OCR systems. To the best of our knowledge, this is the first study to offer a sizable historical dataset with human-level performance in this domain.

In addition to the human-level performance, we also demonstrated the utility of our dataset in tackling text-image recognition challenges. Our evaluation involved the previously proposed smaller-size models for the Amharic script and SOTA models that has been validated with Latin-based and Chinese scripts. Our experiments demonstrate that both the trained SOTA methods and the smaller networks yield comparable results. Notably, the SOTA models produce equivalent outcomes even with smaller iterations but larger parameter size, which shows the potential for smaller networks to be suitable for low-resource computing infrastructure while still achieving comparable results. The dataset and source code can be accessed at `https://github.com/ethopic/hhd-ethiopic-I`. As part of our future work, we plan to expand the dataset and incorporate language models and contextual information to enhance recognition performance. In addition, we aim to refine the baseline models and conduct further experiments to enable a more systematic and conclusive evaluation of the different methods.

## 6 REPRODUCIBILTY STATEMENT

We provide the description about the dataset and the open-source implementation details at `https://github.com/ethopic/hhd-ethiopic-I`. Users can reproduce the results by following the information provided at this link. Furthermore, detailed information about the Ethiopic writing system and data collection procedures can be found in the appendix section of this paper. In addition, the dataset datasheet is provided as supplementary material.

## 7 ETHICAL STATEMENT

The authors do not foresee any negative social impacts resulting from this work.

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

APPENDIX

This appendix comprises three sections: Ethiopic writing system, dataset collection and baseline model training details. The Ethiopic writing system section explores background of Ethiopic script, scripting structure and its historical significance. The dataset collection section outlines the data collection, preprocessing and annotation steps, and statistically information about samples in HHD-Ethiopic dataset. Lastly, the model baseline training process section presents insights into baseline model training strategies and sample results.

# A  ETHIOPIC WRITING SYSTEMS

Ethiopic script is an ancient writing system used primarily in Ethiopia and Eritrea. With its origins dating back to the $4^{th}$ century AD Jeffrey (2004b). The script is characterised by its unique syllabic structure, which combines consonants and vowels to form complex characters. In literature the Ethiopic writing system also named with various names including "Abugida", "Amharic", "Ge'ez", and "Fidel".

Ethiopic script has been a significant cultural and linguistic heritage of the region, playing a vital role in preserving the rich history and traditions of Ethiopia. It is primarily used for writing over 27 languages including the Amharic and Tigrinya languages, among others. As depicted in Figure 8, the script has a distinct visual appearance, characterized by its curved and geometric shapes, making it visually distinctive and is written and read, as English, from left to right and top to down Belay et al. (2019a).

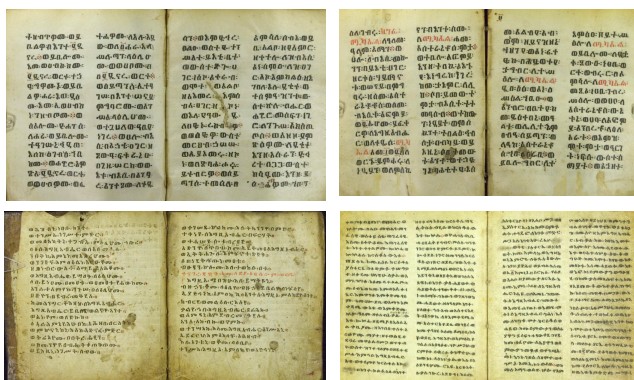

Figure 8: Sample historical handwritten Ethiopic manuscripts

Despite the long history of the Ethiopic script, it has encountered numerous challenges in the digital world due to its low-resource nature Destaw et al. (2022); Seyoum et al. (2016). Issues such as limited digitized fonts, linguistic tools, and datasets have posed obstacles in the fields of natural language processing and document image analysis technologies.

The Ethiopic script poses unique challenges for machine learning due to the scarcity of available resources. This script is characterized by its complex orthographic identities and visually similar characters. Comprising over 317 distinct characters, including approximately 280 characters organized in a 2D matrix format known as Fidel-Gebeta (Figure 9), along with 20 digits and 8 punctuation marks (Figure 10).

As depicted in Figure 9, the Ethiopic script consists of 34 consonant characters, which serve as the base for deriving additional characters using diacritics. These diacritics can be found as small marks placed on the top, bottom, left, or right sides of the base character. Furthermore, specific vowel characters are formed by shortening either the left or right leg of consonant characters, as demonstrated in columns 4 (shortening left leg) and 7 (shortening right leg) of the fidel-Gebeta. The vowels, derived from these consonants, span from 1 to 12 and correspond to the respective columns.

|  |  | 1 | 2 | 3 | 4 | 5 | 6 | 7 | 8 | 9 | 10 | 11 | 12 |
|---|---|---|---|---|---|---|---|---|---|---|---|---|---|
|  |  | ä/e | u | i | a | ē | ə | o | ʷä/ue | ʷi/u | ʷa/ua | ʷē/uē | ʷə |
| 1 | h | ሀ | ሁ | ሂ | ሃ | ሄ | ህ | ሆ |  |  |  |  |  |
| 2 | l | ለ | ሉ | ሊ | ላ | ሌ | ል | ሎ |  |  | ሏ |  |  |
| 3 | ḥ | ሐ | ሑ | ሒ | ሓ | ሔ | ሕ | ሖ |  |  | ሗ |  |  |
| 4 | m | መ | ሙ | ሚ | ማ | ሜ | ም | ሞ |  |  | ሟ |  |  |
| 5 | ś | ሠ | ሡ | ሢ | ሣ | ሤ | ሥ | ሦ |  |  | ሧ |  |  |
| 6 | r | ረ | ሩ | ሪ | ራ | ሬ | ር | ሮ |  |  | ሯ |  |  |
| 7 | s | ሰ | ሱ | ሲ | ሳ | ሴ | ስ | ሶ |  |  | ሷ |  |  |
| 8 | š | ሸ | ሹ | ሺ | ሻ | ሼ | ሽ | ሾ |  |  | ሿ |  |  |
| 9 | q | ቀ | ቁ | ቂ | ቃ | ቄ | ቅ | ቆ | ቈ | ቊ | ቋ | ቌ | ቍ |
| 10 | b | በ | ቡ | ቢ | ባ | ቤ | ብ | ቦ |  |  | ቧ |  |  |
| 11 | v | ቨ | ቩ | ቪ | ቫ | ቬ | ቭ | ቮ |  |  | ቯ |  |  |
| 12 | t | ተ | ቱ | ቲ | ታ | ቴ | ት | ቶ |  |  | ቷ |  |  |
| 13 | č | ቸ | ቹ | ቺ | ቻ | ቼ | ች | ቾ |  |  | ቿ |  |  |
| 14 | ḫ | ኀ | ኁ | ኂ | ኃ | ኄ | ኅ | ኆ | ኈ | ኊ | ኋ | ኌ | ኍ |
| 15 | n | ነ | ኑ | ኒ | ና | ኔ | ን | ኖ |  |  | ኗ |  |  |
| 16 | ñ | ኘ | ኙ | ኚ | ኛ | ኜ | ኝ | ኞ |  |  | ኟ |  |  |
| 17 | ' | አ | ኡ | ኢ | ኣ | ኤ | እ | ኦ |  |  | ኧ |  |  |
| 18 | k | ከ | ኩ | ኪ | ካ | ኬ | ክ | ኮ | ኰ | ኲ | ኳ | ኴ | ኵ |
| 19 | x | ኸ | ኹ | ኺ | ኻ | ኼ | ኽ | ኾ | ዀ | ዂ | ዃ | ዄ | ዅ |
| 20 | w | ወ | ዉ | ዊ | ዋ | ዌ | ው | ዎ |  |  |  |  |  |
| 21 | ' | ዐ | ዑ | ዒ | ዓ | ዔ | ዕ | ዖ |  |  |  |  |  |
| 22 | z | ዘ | ዙ | ዚ | ዛ | ዜ | ዝ | ዞ |  |  | ዟ |  |  |
| 23 | ž | ዠ | ዡ | ዢ | ዣ | ዤ | ዥ | ዦ |  |  | ዧ |  |  |
| 24 | y | የ | ዩ | ዪ | ያ | ዬ | ይ | ዮ |  |  |  |  |  |
| 25 | d | ደ | ዱ | ዲ | ዳ | ዴ | ድ | ዶ |  |  | ዷ |  |  |
| 26 | ǧ | ጀ | ጁ | ጂ | ጃ | ጄ | ጅ | ጆ |  |  | ጇ |  |  |
| 27 | g | ገ | ጉ | ጊ | ጋ | ጌ | ግ | ጎ | ጐ | ጒ | ጓ | ጔ | ጕ |
| 28 | ṭ | ጠ | ጡ | ጢ | ጣ | ጤ | ጥ | ጦ |  |  | ጧ |  |  |
| 29 | č̣ | ጨ | ጩ | ጪ | ጫ | ጬ | ጭ | ጮ |  |  | ጯ |  |  |
| 30 | p | ጰ | ጱ | ጲ | ጳ | ጴ | ጵ | ጶ |  |  | ጷ |  |  |
| 31 | ṣ | ጸ | ጹ | ጺ | ጻ | ጼ | ጽ | ጾ |  |  | ጿ |  |  |
| 32 | ṣ́ | ፀ | ፁ | ፂ | ፃ | ፄ | ፅ | ፆ |  |  |  |  |  |
| 33 | f | ፈ | ፉ | ፊ | ፋ | ፌ | ፍ | ፎ |  |  | ፏ |  |  |
| 34 | **p** | ፐ | ፑ | ፒ | ፓ | ፔ | ፕ | ፖ |  |  | ፗ |  |  |

Figure 9: Fidel-Gebeta: the row-column matrix structure of Ethiopic characters. The first column shows the consonants, while the following columns (1-12) illustrate syllabic variations (obtained by adding diacritics or modifying parts of the consonant).

For example, in the second row of the fidel-Gebeta, the consonant character ለ represents the sound "le" in Ethiopic. From this base character, various vowel characters emerge, such as:

- ሉ is formed by adding a horizontal diacritic at the middle left side of the base character and represents the sound "lu".
- ሊ is formed by adding a horizontal diacritic at the bottom left leg of the base character and represents the sound "li".
- ላ is formed by shortening the left leg of the base character and represents the sound "la".

These examples showcase the versatility of the Ethiopic script, where modifying the diacritics or leg lengths of consonant characters allows for the representation of different vowel sounds.

Ethiopic numerals also called Ge'ez numerals, are a numeric system traditionally used in Ethiopic writing. These numeral system has its own distinct symbols for representing numbers, which are different from the Arabic or Roman numerals commonly used in many other parts of the world. The system has a base of 10, with unique characters for each digit from 1 to 9, as well as special symbols for tens, hundreds, and thousands (Figure 10). For example:

- Ethiopic symbol ፩ is similar to the Arabic numeral 1.
- symbol ፵፬ is similar to the Arabic numeral 44.

| C | ፪ | ፫ | ፬ | ፭ | ፮ | ፯ | ፰ | ፱ | ፲ |
|---|---|---|---|---|---|---|---|---|---|
| 1 | 2 | 3 | 4 | 5 | 6 | 7 | 8 | 9 | 10 |
| ፳ | ፴ | ፵ | ፶ | ፷ | ፸ | ፹ | ፺ | ፻ | ፼ |
| 20 | 30 | 40 | 50 | 60 | 70 | 80 | 90 | 100 | 10000 |

**a**

| ※ | ፡ | ። | ፣ | ፤ | ፥ | ፧ | ፨ |
|---|---|---|---|---|---|---|---|
| section mark | word separator | full stop (period) | comma | semicolon | colon | question mark | paragraph separator |

**b**

Figure 10: Numbering system (a) and punctuation marks (b) in Ethiopic script

- symbol ፱፼፱፻፱፼፱ similar to the Arabic numeral 99999.

- symbol ፼፪ is similar to the Arabic numeral 10002.

- symbol ፲፪፻፴፫ similar to the Arabic numeral 1233.

Though modern Arabic numerals dominate daily life and official documents, understanding Ethiopic numerals is vital for deciphering historical texts and preserving cultural heritage.

In the Ethiopic writing system, punctuation marks convey meaning and guide text interpretation (see Figure 10). Understanding their usage is vital for clear and effective written communication in Ethiopic script.

The complexities of symbols within the Ethiopic script present significant challenges for machine learning tasks, requiring attentive approaches to achieve accurate recognition and analysis. An example of these challenges is the non-standardized usage of punctuation marks 11 and variations in writing styles, as depicted in Figure 8. These factors contribute to the difficulties encountered in the development of Ethiopic OCR systems.

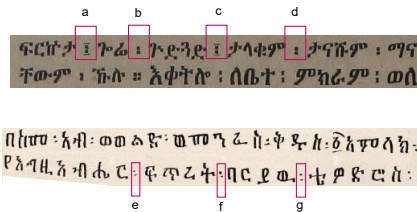

Figure 11: Examples of punctuation usage and writing Styles: As shown by the red rectangle and labeled by [a, b, c, d], there is typically a space before and after the punctuation mark. In contrast, the punctuation marks labeled by [e, f, g] do not have any space before or after them. The punctuation marks labeled by a and c serve as list separators and are distinct from the other punctuation marks, which are used as word separators.

## B    DATA COLLECTION AND ANNOTATION PROCESS

The Ethiopic script, one of the oldest in the world, is underrepresented in the fields of document image analysis (DIA) and natural language processing (NLP). This is due to the lack of attention from researchers in these fields and the absence of annotated datasets suitable for machine learning. However, in recent times, there has been a significant increase in interest from individuals involved in computing and digital humanities. As part of this growing attention, we have contributed by preparing this first sizable historical handwritten dataset for Ethiopic text-image recognition. The primary source of these documents is the Ethiopian National Archive and Library Agency (ENALA), spanning from the $18^{th}$ to the $20^{th}$ century. To ensure privacy, each page is randomly sampled from about seven different books covering cultural and religious related contents. After obtaining scanned

copies of the documents from ENALA, we utilize the OCRopus[2] OCR framework and the ground-truth text annotation process is described as follows:

The annotation process can be grouped in three phase:

- **Phase-I**: In this phase, we hired 14 individuals who are familiar with the Ethiopic script. Out of the 14, 12 were assigned the task of annotation, while the remaining two served as supervisors responsible for follow-up the annotation process and ensuring the completeness of each annotation submission. Additionally, the supervisors were responsible for multiple tasks, including monitoring the progress of each annotator, providing assistance when issues arose, making decisions to address any problems encountered during the annotation process, checking alignment consistency between images and ground-truth at each phase of the annotator's submission, and making necessary corrections in case of errors. Throughout the annotation process, all annotators and supervisors had the freedom to refer to any necessary references.

- **Phase-II**: Once we have all the annotated text-line images from phase-I, we divide the text-image into training and test sets. For the training set, we reserve all text line images from the $19^{th}$ and $20^{th}$ centuries, as well as a few documents with unknown publication dates. The test set is exclusively composed of text line images from the $18^{th}$ century. Additionally, we randomly sample another test set, which constitutes 10% of the training set. We call this randomly selected set as **Test-set-I**, which allows us to evaluate the baseline performance in the classical IID (Independently and Identically Distributed) setting.

  On the other hand, the test set that is drawn from a different distribution than the training set, known as Out-Of-Distribution (OOD), is called **Test-set-II**. This setup enables us to assess the performance in real scenarios where the test set differs from the training distribution.

- **Phase-III**: In this phase, we hired approximately 20 individuals who are familiar with the Ethiopic script, along with one historical expert for the second round of annotation and request them to submit within 5 weeks. This annotation phase has the following two objectives:

  - to ensure the quality of the test set.
  - to evaluate the human-level performance in historical Ethiopic script recognition, which serves as a baseline for comparison with machine learning models.

  Out of the 20 individuals hired, only 13 annotators successfully completed the annotation task within the specified submission deadline, while the remaining individuals failed and resigned from the task. Among the 13 successful annotators, the first group comprised 9 people who transcribed text-line images from the first test set, which consisted of 6,375 randomly selected images from the training set. The second group consisted of 4 people who transcribed the second test set, consisting of 15,935 images from the $18^{th}$ century.

  With the exception of the expert reviewer, who was allowed to use external references, all annotators in this phase were instructed to perform the task without the use of references. Detailed data from each annotator was documented as metadata for future reference and can be accessed from our GitHub repository. One observation we made during this annotation process was that some annotators anonymously shared information, despite our efforts to ensure data confidentiality. However, despite this limitation, we have successfully compute the human-level performance for each annotator and have reported the results accordingly.

Considering the resources available to the annotators, including computing infrastructure and internet access, we developed a simple user-friendly tool with a easy to use Graphical User Interface (GUI) for the annotation process. The tool is depicted in Figure 12 and sample text-line images with the corresponding ground truth are shown in figure 13.

Each annotator's machine was equipped with this tool, enabling them to work offline when internet access was unavailable. Additionally, we provided them with a comprehensive *README* file and instructed them on how to utilize the annotation tool.

---

[2]https://github.com/ocropus/ocropy

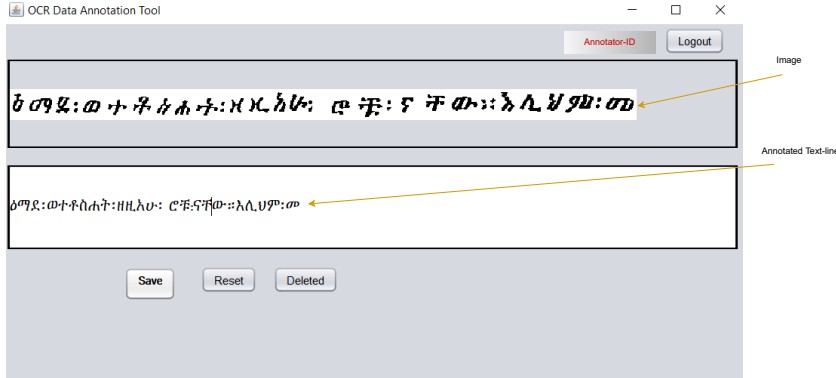

Figure 12: Text-line image annotation tool

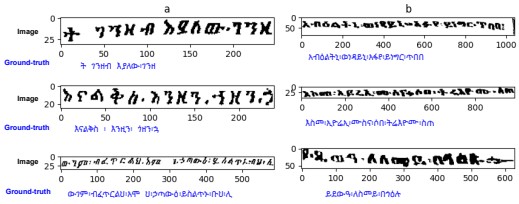

Figure 13: Sample text-line images and ground-truth for HHD-Ethiopic: a) Training-set. b) Test-set.

## B.1 DATASET STATISTICAL OVERVIEW AND COMPARISONS

This section provides a detailed description of the characteristics of the HHD-Ethiopic dataset. These characteristics include the diversity of content, variations in image quality, distribution of image sizes in the trainin and test sets, the number of samples per class, and a comparison with related datasets. Examples of sample page images are illustrated in Figure 14, showcasing pages

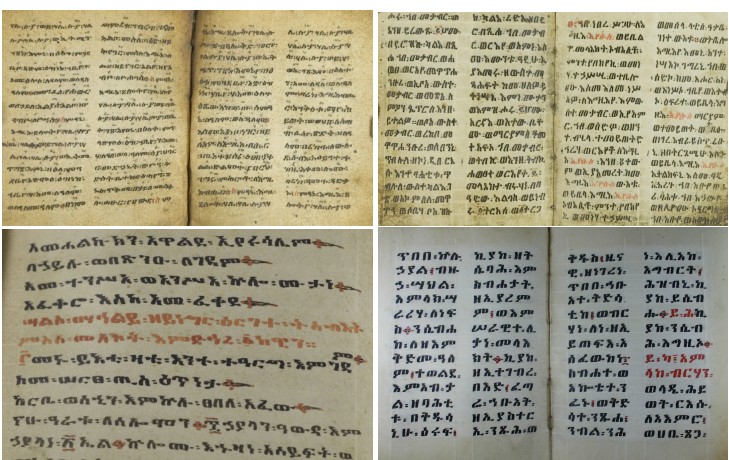

Figure 14: Sample page images ranging from $18^{th}, 19^{th}, 20^{th}$ centuries, as well as images of unknown publication dates, arranged from top left, top right, bottom left and bottom right respectively.

from various publication years (categorized as $18^{th}, 19^{th}, 20^{th}$, and unknown date of publication). In addition, Figure 15 displays page images categorized by image quality, which ranges from bad to medium and good. It's important to note that documents of insufficient quality, falling below the "bad" threshold, are excluded during the process of text line extraction.

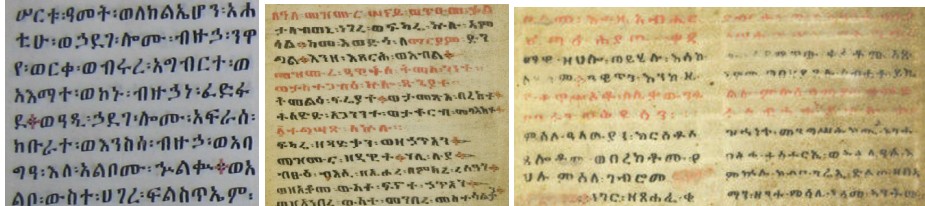

Figure 15: Sample page image images with good(left) , medium(middle) and bad (right) quality.

The histogram in Figure 17 illustrates the distribution of text-line image sizes (width and height) across the training set and two test sets. Additionally, access to the distribution of characters for each class (i.e., the frequency of characters within the 306 unique characters) in both the training and test sets.

To better represent characters that are infrequent or absent in the training set, we have employed a solution involving the generation of synthetic images. Each character is incorporated into synthetic images approximately 200 times on average.



Figure 16: Frequency distribution of underrepresented characters occurring 20 times or less in the training set. zero in the frequency column refers to the characters that exit in the test set but not in the training set.

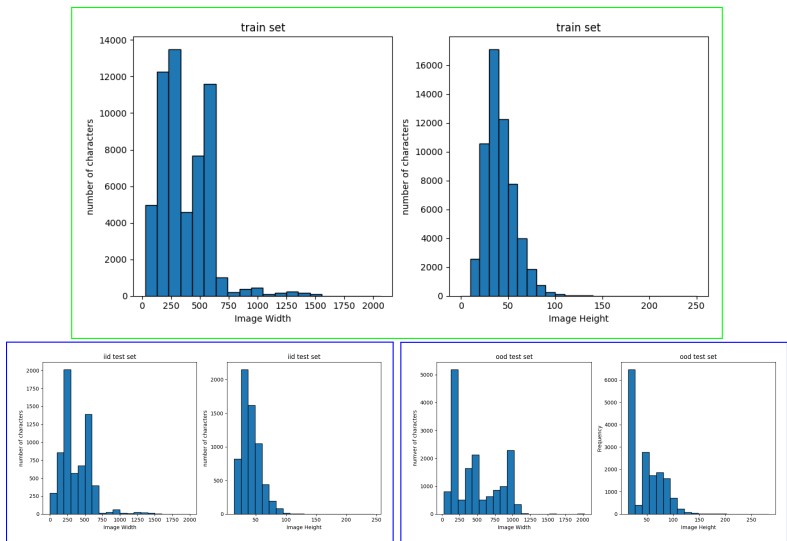

Figure 17: A histogram for distribution of image sizes in the HHD-Ethiopic dataset: a) Training-set (top). b) IID test-set (bottom left), c) OOD test set(bottom right).

Though it may not be fair to directly compare datasets from distinct settings, we provide a comparisons between our historical handwritten (HHD-Ethiopic) dataset and the existing collections of modern printed, modern handwritten, and scene text datasets for the task of Ethiopic script recognition. The summary of comparisons is is given in Table B.1.

Table 4: Summary of publicly available datasets for Ethiopic script

| Dataset-type | image-type | # images | # uniq-chars | # test-sample | annotations |
|---|---|---|---|---|---|
| PrintedBelay et al. (2019a) | real | 40,929 | 280 | 2,907 | line-level |
| | synthetic | 296,408 | 280 | 15724 | line-level |
| SceneDikubab et al. (2022) | real | 15,39 | 302 | 9,257 | word-level |
| | synthetic | 2.8M | 302 | - | word-level |
| HandwrittenAbdurahman et al. (2021) | real/modern | 12,064 | 300 | 1,2064 | word-level |
| | Augmented | 33,672 | - | * | word-level |
| HandwrittenAssabie & Bigun (2011) | real/modern | 10,932 | 265 | - | word-level |
| **Our** (HHD-Ethiopic) | real/historical | 79,684 | 306 | 22,310 | line-level |
| | synthetic | 1200 | 64 | * | line-level |

the synthetic data generated for our new dataset denotes the number of underrepresented characters

- denotes information that is unavailable/ not given
* denotes data that has not been utilized for testing

## C  BASELINE MODELS AND IMPLEMENTATION DETAILS

In this section, we provide additional details of models implemented and evaluated on our HHD-Ethiopic OCR dataset. We evaluate several state-of-the-art methods, which can be broadly grouped as CTC-based, Attention, and Transformer-based. However, our primary focus in this section is on the CTC-based model, which is designed to work effectively in lower resource settings. This is because the other CTC, attention and Transformer-based model ( evaluated on this new datasets) are validated from previous works Du et al. (2022b); Fang et al. (2021b); Shi et al. (2016b; 2018b) and involves larger parameter size, making it more suited for higher-resource environments. These SOTA methods are implemented based on the open-source toolbox, mmocr: `https://github.com/open-mmlab/mmocr.`

### C.1  BASELINE MODELS

The implementation of the CTC-based model follows a typical pipeline depicted in Figure 18. In case of Plain-CTC, initially, the preprocessed images are passed through a convolutional neural network (CNN) backbone, which extracts relevant image features using a series of convolutional and pooling layers.

The output features from the CNN backbone are reshaped and subsequently fed into a a Long Short-term Memory (LSTM) network with connectionist temporal classification (CTC) network. This combination enables the model to effectively capture the temporal dependencies between the image features and the corresponding text labels. The RNN layer incorporates two Bi-directional LSTM units to learn sequential patterns and generate a $[(c+1) \times T]$ matrix of Softmax probabilities for each character at each time-step, where c and T denote the number of characters and the length of maximum time-step. Finally, a the CTC converts the intermediate representations into the final output text predictions.

The alternative CTC-based approach, referred to as Attn-CTC within this paper and previously introduced for Amharic text recognitionBelay et al. (2021b), extends the Plain-CTC methodology by incorporating an attention mechanism into the CTC layers. The rationale behind incorporating the attention layer lies in leveraging its capacity to derive a more potent hidden representation through a weighted contextual vector. This model comprises a combination of CNN and LSTM as the encoding module. The output of this module feeds into the attention module, and subsequently, the decoded output string is obtained through the CTC layer.

During training, the CTC algorithm calculates the likelihood of the output sequence given the input sequence and uses it as the objective function Graves et al. (2006b); Liwicki et al. (2007). The training process maximizes this likelihood, which, in turn, maximizes the probability of the correct output sequence. The loss that is minimized during training is the negative of this likelihood, which can be defined as:

$$CTC_{loss} = -log \sum_{(y,x) \in S} p(y/x) \qquad (1)$$

where x and y denote pair of input and output sequences in sample dataset S respectively and the probability of label sequence for a single pair p(y/x) is computed by multiplying the probability of labels along a specific path $\pi$ for the overall time steps T and it can be defined as:

$$P(y/x) = \prod_{t=1}^{T} p(a_t, \pi) \qquad (2)$$

where a is a character in the specified path and p(a) is its probability on each time-step on that path.

Once training and evaluating the OCR model with network settings proposed in Belay et al. (2020b; 2021b), we employed Bayesian optimization for the selection of hyperparameters, with the CTC validation loss serving as the criteria for optimization. Bayesian optimization captures the relationship between the hyperparameters and the CTC validation loss, iteratively updating and refining the model as it explores different hyperparameter configurations ( see ref Balaprakash et al. (2018b)for details) that yields lower CTC validation loss values. This approach allowed us to effectively tune our model and enhance its performance, contributing to the overall success of our text-image recognition model.

The recognition performance of all human-level and baseline models evaluated in this work is reported using the character error rate (CER) and Normalized Edit Distance (NED) metrics. All results reported with these two metrics are converted to 100%. The CER metric can be computed as follows,

$$CER(T, P) = \left( \frac{1}{c} \sum_{m \in T, n \in P} ED(m, n) \right) \times 100, \qquad (3)$$

where $c$ denotes the total number of characters in the ground-truth, $t$ and $p$ denote the ground-truth labels and predicted respectively, and $ED(m, n)$ is the Levenshtein edit-distance between sequences $m$ and $n$.

while the NED metric is computed as:

$$NED = \left( \frac{1}{N} \sum_{i=1}^{N} \frac{ED(m_i, n_i)}{\max(l_i, \hat{l}_i)} \right) \times 100 \qquad (4)$$

where N is the maximum number of paired ground truth and prediction strings, $ED$ is the Levenshtein edit distance, $m_i$ and $n_i$ denote the predicted text and the corresponding ground truth (GT) string, respectively, and $l_i$ and $\hat{l}_i$ are their respective text lengths.

## C.2 TRAINING DETAILS AND CONFIGURATIONS

During our experiments, we employed various hyperparameter settings, including those selected by Bayesian Optimization Balaprakash et al. (2018b) specifically for the CTC-based models. Training and evaluation were performed on a single NVIDIA RTX A6000 GPU for all the baseline models.

For the CTC-based baseline models, we trained them multiple times with different hyperparameter values, including epochs ranging from 10 to 100, employing a trial-and-error approach and utilizing the hyperparameters suggested by Bayesian Optimization. In this paper, we report the results obtained from the two CTC-based models (without attention) achieving better CER in 15 epochs. Additionally, the attention-CTC models showed improved performance as we trained them for more epochs. The reported results, for attention-CTC models, in the main paper were trained for 100 epochs. The dataset and the code can be accessed at `https://github.com/ethopic/hhd-ethiopic-I`

Considering our focus on low-resource settings, we prioritize optimizing our time and resources effectively. Hence, as it is not suitable for training in resource-constrained environments, we do not

---

[6]https://deephyper.readthedocs.io/en/latest/index.html

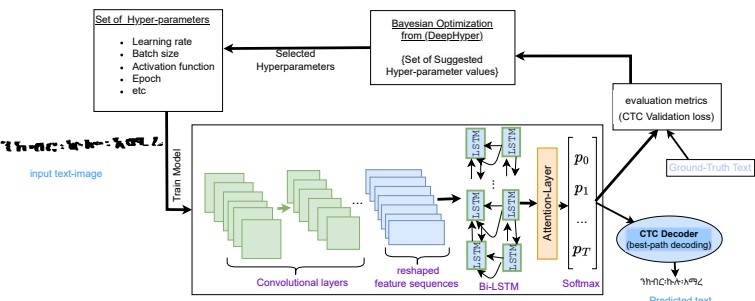

Figure 18: A typical view of the proposed model and set of best hyper-parameters value selection using Bayesian optimization from DeepHyper[6]. The output denoted by $p_0, p_1...p_T$, is a matrix of Softmax probabilities with dimensions $[(c+1) \times T]$, where c is the number of unique characters in the ground-truth text and T is the length of the input time-step to the LSTM layers. The validation loss was utilized as the metric for tuning the hyperparameters. To obtain the final output sequence from the predicted probabilities produced by the model, we use the best-path decoding strategy.

recommend utilizing complex models for Ethiopic text recognition. Instead, we prioritize exploring alternative models ( such as the smaller CTC-based methods discussed in the main paper) which balance between computational efficiency and performance to ensure the feasibility of the OCR system in limited resources. However, if you possess significant computing resources, using synthetic data and conducting more extensive training iterations on those models could lead to an improvement in recognition performance for historical handwritten Ethiopic manuscripts.

We evaluated various SOTA models Du et al. (2022b); Fang et al. (2021b); Shi et al. (2016b; 2018b) using our HHD-Ethiopic dataset. Although these models still have a relatively high number of parameters in comparison to the CTC-based models ( the plain and Attn-CTC), they remain more manageable in low-resource settings. Despite the increased parameter count, we run these models for 25 epochs using limited computational resources. We achieved an improved recognition performance compared to the results presented in the TrOCR paper. By balancing performance and resource demands, the models Du et al. (2022b); Fang et al. (2021b); Shi et al. (2016b; 2018b) present a viable option for practical deployment and utilization, especially in situations where computational resources are constrained.

Due to the limited number of experimental runs conducted for Du et al. (2022b); Fang et al. (2021b); Li et al. (2023); Shi et al. (2016b; 2018b) baseline models, we decided not to include box plots for all baseline models in the main paper. Box plots are commonly used to visualize results distribution across multiple runs, allowing for the assessment of variations and identification of outliers. Since a box plot is not suitable for representing a single experiment, we have illustrated the learning curve of the four models (ABINet, ASTER, SVTR and CRNN) in Figure 19. This learning curve illustrates the recognition performance on both IID and OOD test sets using the CER and metric across 25 epochs.

Based on learning curve depicted in Figure 19, we can conclude that all models would perform better as we train for longer epochs. Within the first 25 epochs, SVTR outperforms the others, while ASTER is the least performer. We are limited to running for 25 epochs due to time and computational resources. The red horizontal line in both the right and left plots represents the CER for Hopt-attn-CTC model. This line serves as our benchmark, as it represents the best-performing model.

---

[1] Please note that the CER can exceed 100% when the predicted text is much longer than the ground truth. Excessive length leads to an edit distance surpassing the ground truth's character count. For instance, if the ground truth is 'ab' and the prediction is 'abced' the edit distance is 3 compared to the ground truth's 2 characters. This results in a ratio of 1.5*100=150 (see equations 3). In contrast, NED ranges from 0 to 100%, where values close to 0 are better, while values closer to 100% are indicative of poorer performances in both metrics.

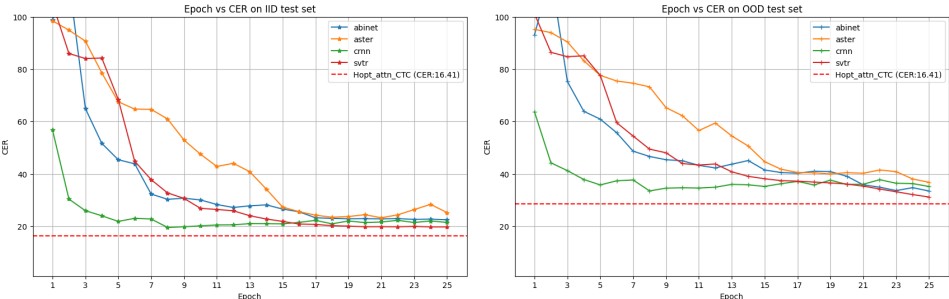

Figure 19: Learning curve on IID and OOD test data. CER[1] on IID test set (left), CER on OOD test set (right) across 25 epochs for ASTER, ABInet, SVTR, and CRNN models. In all plots, the red horizontal line represents the CER value of the Hopt-attn-CTC network on IID and OOD data respectively.

## C.3 SAMPLE PREDICTED TEXTS

Sample images with the corresponding ground truth, model prediction and the edit distance between the ground truth and the prediction at line level is shown in Figure20

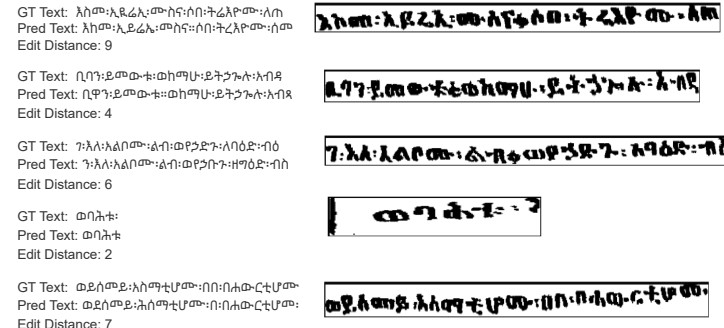

Figure 20: Sample text-line images with their corresponding ground-truth and prediction texts

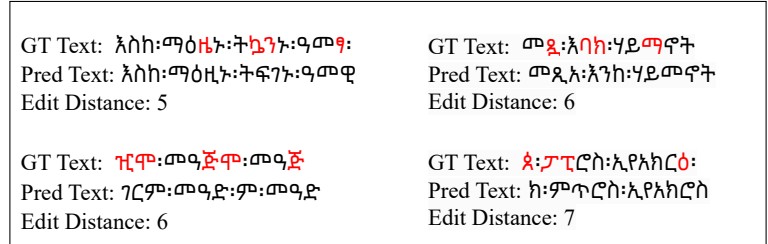

Figure 21: Examples of prediction errors for underrepresented characters. The characters marked in red within the ground-truth text are less frequent characters and are wrongly predicted.

In text lines where characters with low occurrence rates appear in the ground truth of the training set often leads to an increased edit distance between the ground truth and the predicted texts during test time. This pattern is demonstrated by sample examples depicted in Figure.21

