# Supplementary file for "HHD-Ethiopic: A Historical Handwritten Dataset for Ethiopic OCR"

## Introduction

This document comprises three sections: dataset documentation (datasheet for a dataset)

## A    Dataset documentation for HHD-Ethiopic

To prepare this dataset documentation, we use a datasheet [1] for dataset guideline. This documentation consists of the motivation behind the dataset, its composition, the process of collection, recommended use cases, as well as information on processing, cleaning, labeling, distribution (including hosting, licensing), and maintenance. This documentation also includes author statements.

### A.1    Motivation

**For what purpose was the dataset created?** Was there a specific task in mind? Was there a specific gap that needed to be filled? Please provide a description.

The dataset targets the challenges of the indigenous Ethiopic script, addressing its scarcity of resources. It serves as a valuable asset for researchers and developers, facilitating advancements in OCR technology specifically for historical handwritten Ethiopic recognition. Unlike well-studied scripts like Latin, it bridges the gap and enables accurate recognition of Ethiopic text in historical documents using machine learning approaches.

**Who created this dataset** (e.g., which team, research group) and on behalf of which entity (e.g., company, institution, organization)?

We will include the the data creator after the paper review for the sake of anonymity..

**Who funded the creation of the dataset?** If there is an associated grant, please provide the name of the grantor and the grant name and number.

We will include the funding agencies after the paper review for the sake of anonymity..   **Any other comments?** No.

### A.2    Composition

**What do the instances that comprise the dataset represent (e.g., documents, photos, people, countries)?** Are there multiple types of instances (e.g., movies, users, and ratings; people and interactions between them; nodes and edges)? Please provide a description.

The HHD-Ethiopic dataset is an OCR dataset that consists of text-line images extracted from historical handwritten Ethiopic manuscript and there corresponding ground truths text.

**How many instances are there in total (of each type, if appropriate)?**

The HHD-Ethiopic dataset comprises 79,684 text-line images accompanied by their respective ground-truth texts. These images are extracted from a collection of 1,746 pages of Ethiopic manuscripts

dating from the $18^{th}$ to the $20^{th}$ centuries. The dataset is divided into a training set, containing 57,374 text-line images, and two distinct Test sets. One Test set, that consists 6,375 images, is randomly sampled from the training set, while the other is exclusively prepared from the $18^{th}$ century Ethiopic manuscripts and includes about 15,935 text-line images along with their corresponding ground-truth texts ( details are provided in the main paper).

**Does the dataset contain all possible instances or is it a sample (not necessarily random) of instances from a larger set?** If the dataset is a sample, then what is the larger set? Is the sample representative of the larger set (e.g., geographic coverage)? If so, please describe how this representativeness was validated/verified. If it is not representative of the larger set, please describe why not (e.g., to cover a more diverse range of instances, because instances were withheld or unavailable).

HHD-Ethiopic, is a historical handwritten dataset between the $18^{th}$ and $20^{th}$ centuries. It is a sample of instances from that time period and includes 306 out of 317 frequently used characters in the Ethiopian writing system.

**What data does each instance consist of? "Raw" data (e.g., unprocessed text or images) or features?** In either case, please provide a description.

Each instance in the training set consists of text-line images and their corresponding ground-truth text. The test set, on the other hand, includes raw human-level prediction texts from 13 independent annotators which we use as a baseline to compare the human-level performance with OCR models in this paper

**Is there a label or target associated with each instance?** If so, please provide a description.

Yes, there is a ground-truth text for each text-line image.

**Is any information missing from individual instances?** If so, please provide a description, explaining why this information is missing (e.g., because it was unavailable). This does not include intentionally removed information, but might include, e.g., redacted text.

No, everything is included.

**Are relationships between individual instances made explicit (e.g., users' movie ratings, social network links)?** If so, please describe how these relationships are made explicit.

The relationships between individual instances in the text-line image dataset are not explicitly defined, as each image is formed from a sampled set of 306 Ethiopic characters rather it may have indirect/inferred connection.

**Are there recommended data splits (e.g., training, development/validation, testing)?** If so, please provide a description of these splits, explaining the rationale behind them.

The HHD-Ethiopic dataset is split into first into training, and testing. The training set includes text-line images from the $19^{th}$ and $20^{th}$ centuries. A validation set is then randomly sampled as 10% of the training set. Two test sets are propose: the first testing set consists of 6,375 images randomly selected from a similar distribution as the training set. The second testing set contains 15,935 images from a different distribution, representing $18^{th}$ century manuscripts. The first test evaluates baseline performance in an IID setting, while the second test assesses performance in an OOD scenario. The detail statistic is provided in section 3 of the main paper.

**Are there any errors, sources of noise, or redundancies in the dataset?** If so, please provide a description.

While the ground-truth text was double-checked by a supervisor for each annotator, we recommend additional revision of the the ground-truth texts by multiple historical document experts to minimize annotation errors.

**Is the dataset self-contained, or does it link to or otherwise rely on external resources (e.g., websites, tweets, other datasets)?** If it links to or relies on external resources, a) are there guarantees that they will exist, and remain constant, over time; b) are there official archival versions of the complete dataset (i.e., including the external resources as they existed at the time the dataset was created); c) are there any restrictions (e.g., licenses, fees) associated with any of the external resources that might apply to a future user? Please provide descriptions of all external resources and any restrictions associated with them, as well as links or other access points, as appropriate.

The dataset is entirely self-contained. It will exist, and remain constant, over time once we release it.

**Does the dataset contain data that might be considered confidential (e.g., data that is protected by legal privilege or by doctor-patient confidentiality, data that includes the content of individuals non-public communications)?** If so, please provide a description.

No.

**Does the dataset contain data that, if viewed directly, might be offensive, insulting, threatening, or might otherwise cause anxiety?** If so, please describe why.

No.

**Does the dataset relate to people?** If not, you may skip the remaining questions in this section.

No.

**Does the dataset identify any subpopulations (e.g., by age, gender)?** If so, please describe how these subpopulations are identified and provide a description of their respective distributions within the dataset.

No.

**Is it possible to identify individuals (i.e., one or more natural persons), either directly or indirectly (i.e., in combination with other data) from the dataset?** If so, please describe how.

No.

**Does the dataset contain data that might be considered sensitive in any way (e.g., data that reveals racial or ethnic origins, sexual orientations, religious beliefs, political opinions or union memberships, or locations; financial or health data; biometric or genetic data; forms of government identification, such as social security numbers; criminal history)?** If so, please provide a description.

No.

**Any other comments?** No.

## A.3   Collection Process

**How was the data associated with each instance acquired?** Was the data directly observable (e.g., raw text, movie ratings), reported by subjects (e.g., survey responses), or

indirectly inferred/derived from other data (e.g., part-of-speech tags, model-based guesses for age or language)? If data was reported by subjects or indirectly inferred/derived from other data, was the data validated/verified? If so, please describe how.

The historical Ethiopic manuscripts were solely collected from Ethiopian national Archive and Library Agency (ENALA). Each instance is an image/scanned version of documents and is directly observable (see the main paper from section 3).

**What mechanisms or procedures were used to collect the data (e.g., hardware apparatus or sensor, manual human curation, software program, software API)?** How were these mechanisms or procedures validated?

After obtaining the scanned copy of the manuscript from ENALA and extracting the text-image lines, we hire individuals to annotate each text-line image. During the annotation process, all annotators have the freedom to refer to any external sources. for annotation purpose, annotation, we develop an offline tool that can be easily installed on each user's machine and can be accessed `https://github.com/ethopic/hhd-ethiopic-I/tree/main/labeling_tool`.

**If the dataset is a sample from a larger set, what was the sampling strategy (e.g., deterministic, probabilistic with specific sampling probabilities)?**

The historical documents were collected from ENALA. While we did not have the authority to select specific documents, the workers randomly select pages, taking into account our request and the need to maintain the confidentiality of the book's information.

**Who was involved in the data collection process (e.g., students, crowdworkers, contractors) and how were they compensated (e.g., how much were crowdworkers paid)?**

the participants were students and staff members and for the raw manuscript collection and digitization we paid money as a compensation.

**Over what timeframe was the data collected? Does this timeframe match the creation timeframe of the data associated with the instances (e.g., recent crawl of old news articles)?** If not, please describe the timeframe in which the data associated with the instances was created.

The dataset was collected in March-May 2022 and the complete data creation (including preprocessing, annotation and verification were done from September 2022-February 2023.

**Were any ethical review processes conducted (e.g., by an institutional review board)?** If so, please provide a description of these review processes, including the outcomes, as well as a link or other access point to any supporting documentation.

No.

**Does the dataset relate to people?** If not, you may skip the remaining questions in this section.

No.

**Did you collect the data from the individuals in question directly, or obtain it via third parties or other sources (e.g., websites)?**

As described section 3 of the main paper, the data was collected from ENALA directly.

**Were the individuals in question notified about the data collection?** If so, please describe (or show with screenshots or other information) how notice was provided, and provide a link or other access point to, or otherwise reproduce, the exact language of the notification itself.

Yes, the scanned copies of document images were collected directly from ENALA. This request was made in person along with a letter, which also explained the objectives, goals, and the need for data in our work.

**Did the individuals in question consent to the collection and use of their data?** If so, please describe (or show with screenshots or other information) how consent was requested and provided, and provide a link or other access point to, or otherwise reproduce, the exact language to which the individuals consented.

Yes, once we met with the staff at ENALA and explained the goals of our project, they agreed to provide the data and arranged a way for delivering the documents.

**If consent was obtained, were the consenting individuals provided with a mechanism to revoke their consent in the future or for certain uses?** If so, please provide a description, as well as a link or other access point to the mechanism (if appropriate).

No.

**Has an analysis of the potential impact of the dataset and its use on data subjects (e.g., a data protection impact analysis) been conducted?** If so, please provide a description of this analysis, including the outcomes, as well as a link or other access point to any supporting documentation.

No.

### A.4 Preprocessing/cleaning/labeling

**Was any preprocessing/cleaning/labeling of the data done (e.g., discretization or bucketing, tokenization, part-of-speech tagging, SIFT feature extraction, removal of instances, processing of missing values)?** If so, please provide a description. If not, you may skip the remainder of the questions in this section.

Yes, preprocessing tasks such as image segmentation and the removal of non-Ethiopic characters were performed. Furthermore, alignments between the images and their corresponding text-line images were double-checked for each submission by the annotators and verified by a reviewer.

**Was the "raw" data saved in addition to the preprocessed/cleaned/labeled data (e.g., to support unanticipated future uses)?** If so, please provide a link or other access point to the "raw" data.

No.

**Is the software used to preprocess/clean/label the instances available?** If so, please provide a link or other access point.

Yes, here is the link for the labeling tool that we developed with the aim of fitting and making it easier for the target annotators. It is designed to accommodate their operating systems and internet service settings, allowing them to work offline when there is no internet connection. You can access the tool at this link: `https://github.com/ethopic/hhd-ethiopic-I/tree/main/labeling_tool`. For preprocessing tasks, including column detection, binarization, and text-line segmentation, we utilize the OCRopus framework. You can find more information about the framework and its functionalities on their GitHub page: `https://github.com/ocropus/ocropy`

### A.5 Uses

**Has the dataset been used for any tasks already?** If so, please provide a description.

HHD-Ethiopic is a new historical handwritten Ethiopic OCR dataset for a text-line image recognition. In this work we evaluate several state-of-the-art deep learning models and an independent human-level recognition performance on a dataset, which involves comparing the performance of several human annotators with the performance of machine models. The human-level performance serves as a benchmark and in turn it also contribute to the uniqueness and quality of the dataset.

**Is there a repository that links to any or all papers or systems that use the dataset?** If so, please provide a link or other access point.

Yes, we release our dataset, code, baseline models and human-level performances at `https://github.com/ethopic/hhd-ethiopic-I`.

**What (other) tasks could the dataset be used for?**

The HHD-Ethiopic dataset was specifically created to address the gap in Historical handwritten Ethiopic manuscript recognition. However, it can also be utilized to benchmark the performance of machine learning models for other scripts.

**Is there anything about the composition of the dataset or the way it was collected and preprocessed/cleaned/labeled that might impact future uses?** For example, is there anything that a future user might need to know to avoid uses that could result in unfair treatment of individuals or groups (e.g., stereotyping, quality of service issues) or other undesirable harms (e.g., financial harms, legal risks) If so, please provide a description. Is there anything a future user could do to mitigate these undesirable harms?

The datasets can be used without further considerations.

**Are there tasks for which the dataset should not be used?** If so, please provide a description.

No.

**Any other comments?** No.

### A.6 Distribution

**Will the dataset be distributed to third parties outside of the entity (e.g., company, institution, organization) on behalf of which the dataset was created?** If so, please provide a description.

Yes, both the dataset and baseline results will be made available to the public research community for experimentation and further work on historical handwritten recognition.

**How will the dataset will be distributed (e.g., tarball on website, API, GitHub)** Does the dataset have a digital object identifier (DOI)?

The HHD-Ethiopic dataset can be downloaded from `https://github.com/ethopic/hhd-ethiopic-I` or directly for the Huggingface `https://huggingface.co/datasets/OCR-Ethiopic/HHD-Ethiopic`. The images can be downloaded as a zipped file. The digital object identifie (DOI) of the dataset is: doi:10.57967/hf/0691. Our dataset has also been made public on Zenodo.org. However, we have chosen to provide it on Hugging Face and GitHub as well, as we believe these platforms are commonly used within the document image analysis and machine learning community.

**When will the dataset be distributed?**

The dataset is currently available for use in our repository.

**Will the dataset be distributed under a copyright or other intellectual property (IP) license, and/or under applicable terms of use (ToU)?** If so, please describe this license and/or ToU, and provide a link or other access point to, or otherwise reproduce, any relevant licensing terms or ToU, as well as any fees associated with these restrictions. This work is licensed under a CC-BY-4.0 International License and available at: `https://github.com/ethopic/hhd-ethiopic-I` or can be directly downloaded from `https://huggingface.co/datasets/OCR-Ethiopic/HHD-Ethiopic`

**Have any third parties imposed IP-based or other restrictions on the data associated with the instances?** If so, please describe these restrictions, and provide a link or other access point to, or otherwise reproduce, any relevant licensing terms, as well as any fees associated with these restrictions.

No.

**Do any export controls or other regulatory restrictions apply to the dataset or to individual instances?** If so, please describe these restrictions, and provide a link or other access point to, or otherwise reproduce, any supporting documentation.

### A.7 Maintenance

**Who will be supporting/hosting/maintaining the dataset?**

The authors of this paper are responsible for supporting the datasets.

**How can the owner/curator/manager of the dataset be contacted (e.g., email address)?**

The curators of the dataset can be contacted via email and we provide it in the repository `https://github.com/ethopic/hhd-ethiopic-I`

**Is there an erratum?** If so, please provide a link or other access point.

There is no an explicit erratum.

**Will the dataset be updated (e.g., to correct labeling errors, add new instances, delete instances)?** If so, please describe how often, by whom, and how updates will be communicated to users (e.g., mailing list, GitHub)?

Yes, we have plans to add more data to the dataset. As updates are made, we will ensure that both the documentation and our repository are updated accordingly.

**If the dataset relates to people, are there applicable limits on the retention of the data associated with the instances (e.g., were individuals in question told that their data would be retained for a fixed period of time and then deleted)?** If so, please describe these limits and explain how they will be enforced.

No.

**Will older versions of the dataset continue to be supported/hosted/maintained?** If so, please describe how. If not, please describe how its obsolescence will be communicated to users.

Any changes made to the dataset will ensure that the original version remains available, and subsequent versions, such as HHD-Ethiopic-1.1, will be released with documentation.

**If others want to extend/augment/build on/contribute to the dataset, is there a mechanism for them to do so?** If so, please provide a description. Will these contributions be validated/verified? If so, please describe how. If not, why not? Is there a process for communicating/distributing these contributions to other users? If so, please provide a description.

Yes, users can contribute to the dataset and can contact the original authors about incorporating fixes/extensions. This is encouraged. Users are free to extend or augment the dataset for their purposes.

**Any other comments?** None.

### A.8 Accessibility

1. Links to access the **dataset** and its **metadata** and **code and simulation environment**. `https://github.com/ethopic/hhd-ethiopic-I`

2. **Data format**: we follow widely used data formats in OCR dataset. The actual text-line images are stored in .png format wile ground-truth texts are in .txt. the image-ground truth pair are given in .CSV formats, in addition, the images and their corresponding ground-truth are also stored in numpy format. An example of the dataset structure can be found in the README.md file of our dataset repository.

3. **Long-term preservation**: we the authors are responsible to maintain and ensure consistency of the data and it will be in our GitHub repository.

4. **Explicit license**: The dataset is licensed under a CC-BY-4.0 and the source code is under MIT license `https://github.com/ethopic/hhd-ethiopic-I`

5. **A persistent dereferenceable identifier**: A DOI from Hugging Face, doi:10.57967/hf/0691

### A.9 Author statement

The authors have conducted a thorough review of the information presented in this document. To the best of our knowledge, the datasets included in HHD-Ethiopic are intended for research purposes and should be used in accordance with the described methodology and licenses outlined in the Accessibility section. It is important to note that the authors assume full responsibility in the event of any violation of rights.

# Bibliography

[1] Timnit Gebru, Jamie Morgenstern, Briana Vecchione, Jennifer Wortman Vaughan, Hanna Wallach, Hal Daumé Iii, and Kate Crawford. Datasheets for datasets. *Communications of the ACM*, 64(12):86–92, 2021.