# OpenReview forum: "HHD-Ethiopic: A Historical Handwritten Dataset for Ethiopic OCR with Baseline Models and Human-level Performance"
_ICLR.cc/2024/Conference — Submitted to ICLR 2024_

### Official Review · Reviewer_tpDS · 2023-10-25

**Soundness:** 2 fair
**Presentation:** 2 fair
**Contribution:** 1 poor
**Rating:** 3
**Confidence:** 4

**Summary:**

This paper introduces a new dataset and baseline models for historical handwritten Ethiopic OCR. The contributions of this paper are as follows:
(1) A new dataset that contains about 80,000 text-line images from 18th to 20th century manuscripts, with multiple annotations and human-level performance benchmarks.
(2) Three types of classical text recognition methods, including transformer-based methods, attention-based methods, and CTC-based methods, are tested on the new dataset.
(3) The authors compare the baseline methods with human-level performance, showing their superiorities and weaknesses.

**Strengths:**

1. This paper introduces a new dataset for historical handwritten Ethiopic OCR.
2. Some baseline methods are evaluated on this dataset.

**Weaknesses:**

The contributions of this paper are limited. The reasons are:
1. This paper is a dataset paper, without new technical methods.
2. The academic challenges of this dataset are not representative. The impact of this new benchmark seems limited.

**Questions:**

I suggest the authors provide a new solution for this new dataset.

---

> ### Author Response · Authors · 2023-11-18
> **Response to reviewer  tpDS**
>
> Thank you for your comments.
>
> [**This paper is a dataset paper, without new technical methods**]:  Yes, the  paper is a datasets paper and our goal is to  create a historical handwritten dataset for a low-resourced script and provide benchmark results with SOTA OCR  models. In addition we compare the human-machine performance based on evaluation results from several human annotators and the baseline models and we believe this contribution is a good fit for “**dataset and benchmark**" themes (https://iclr.cc/Conferences/2024/CallForPapers) of ICLR-2024.
>
> [**The academic challenges of this dataset are not representative**]: The experimental evaluation results of this dataset, involving several SOTA OCR models and human performance, indicate limited recognition performance. Therefore, this dataset presents challenges and can be valuable for further investigation of new methods to address such challenges. If you have a particular aspect you are curious about, let us know.
>
>
> [**The impact of this new benchmark seems limited**]: To the best of our knowledge, this is the first sizable historical handwritten dataset of the Ethiopic script, characterized by a unique syllabic writing system, low-resource availability, and complex orthographic diacritics. We believe that the human-level performance and results of the SOTA OCR models reported in this work can be utilized as benchmarks for various state-of-the-art OCR models.

---

### Official Review · Reviewer_6Yog · 2023-10-30

**Soundness:** 2 fair
**Presentation:** 2 fair
**Contribution:** 1 poor
**Rating:** 3
**Confidence:** 5

**Summary:**

The article presents a new database of lines of handwritten text from historical documents written in Ethiopian script. This database consists of more than 80,000 lines of text from 1,700 pages. Two test sets were defined, one set with the same distribution as the training set (IID test) and another derived from completely disjoint manuscripts (OOD test). The performance of human transcribers was evaluated. Several handwriting recognition models were trained and evaluated on the test sets. One model outperformed humans on the IID set. The database and training codes are available online.

**Strengths:**

- a large new database available for the Ethiopian script
- the code is available to reproduce the experiments
- an interesting proposal for an IID and OOD test sets

**Weaknesses:**

- full pages are not provides, which prevents the evaluation of full page models, which are currently the best performing models, compared with cascade models (line detection + HTR), which accumulate errors
- no reference to https://journals.openedition.org/jtei/4109
- no reference or comparison to the models available in Transkribus https://readcoop.eu/model/ethiopic-classical-ethiopic-scripts-from-ethiopia-and-eritrea/
- tesseract supports Ethiopic but is neither mentioned nor tested. As the script is not cursive, a test of tesseract would be possible.
- the models tested have never been evaluated on standard handwriting recognition bases. How do they compare with standard libraries such as Transkribus, pylaia and trOCR?

In conclusion, the article presents an interesting resource for HTR, but makes no new contribution either experimentally or methodologically.

**Questions:**

- no measure of inter-annotator agreement: it seems that annotation is difficult, judging by the poor performance of humans. How correct is the annotation? What is the variance of the annotation?
- no details are given about the manuscripts: where do they come from, how were they chosen, how many are there?

---

> ### Author Response · Authors · 2023-11-18
> **Response to reviewer 6Yog**
>
> Thank you for reviewing our work. Your input has helped us improve the paper. We've addressed your questions and comments below.
>
> [**full page are not provided to evaluate line detection+HTR**]: line detection + recognition is one way of OCRing. in literature, the cascaded detection and recognition task  is most common in scene text recognition. Our focus was to create segmented text-line images  for the recognition task and the annotation was done for lines images. However, we will keep updating the dataset for detection and recognition tasks in the future.
>
> [**include reference paper done for Ethiopic**]: we included the paper in our reference list.
>
> [**no reference or comparison with the already available Ethiopic_Eritrea  HTR model in Transkribus**] This model is released in the form of webApp, and cannot be retrained. Furthermore, the authors do not provide an API to easily upload samples and download results. Therefore, we limited ourselves to evaluating its performance manually by uploading a few with a samples manuscript image from our test dataset and comparing the results with those of our model. The Ethiopic_Eritrea  HTR model from the Transkribus model archives a CER of~65.2%  while our Hopt-attn-CTC model archives 17.82 for this sample specific test manuscript. Link for demo:https://github.com/ethopic/hhd-ethiopic-I/blob/main/Hopt_attn_CTC_vs_Transkribu.ipynb
>
> [**the models tested have never been evaluated on standard handwriting recognition bases. How do they compare with standard libraries such as Transkribus, pylaia and trOCR?**]
> We have assessed eight methods, including SOTA OCR techniques validated on Latin, Chinese, and Amharic scripts as documented in the literature, under the same computational constraints and resources. Our focus is on models that are comparatively more lightweight than those requiring extensive computing resources or subscriptions. Additionally, we have incorporated the evaluation results of TrOCR (refer to Table 3 in the revised version). TrOCR, being a model with a large number of parameters, is fine-tuned for fewer epochs. This approach is not chosen because it requires more time for training and inference, leading us to prioritize smaller, more efficient networks. Similarly, the Ethiopic_Eritrea Handwritten Text Recognition (HTR) model in Transkribus, which you recommended and we have evaluated, is developed using PyLaia. It's important to note that models in Transkribus are not free and require a subscription for extensive HTR tasks. They also require a stable internet connection. Furthermore, the need to upload manuscripts for the Transkribus models raises ethical concerns.
>
> [**Tesseract model**], Tesseract is an OCR framework designed for hundreds of scripts including Ethiopic. We evaluated the model proposed for Amharic on a sample test set. The results being so poor, this model cannot be considered state of the art. For this reason we did not include it as part of our baselines.
>
> [**inter-annotator agreement of annotators**]  As discussed in section 3.1 and Appendix B, each text-line image in the training set is annotated individually. In contrast, the test sets undergo expert review, and we evaluate the ground-truth text differences before and after the review in CER (see section 4.1.)
>
> [**details about the manuscripts**] As discussed in appendix B, the text line images are extracted from manuscripts between the 18th and 20th century; collected from ENALA covering 7 different books of cultural and religious contents (~ 1300 pages), while the remaining is from the internet.

---

### Official Review · Reviewer_HsR1 · 2023-11-01

**Soundness:** 2 fair
**Presentation:** 2 fair
**Contribution:** 3 good
**Rating:** 5
**Confidence:** 5

**Summary:**

This paper works on historical handwritten Ethiopic script recognition. It provides introduction to the character system contained in the Ethiopic and lists some major challenges in this task at first. Then, it introduces a new OCR dataset for historical handwritten Ethiopic script HHD-Ethiopic, which consists of roughly 80,000 annotated text-line images and compares the human-level recognition performance with some state-of-the-art OCR models on the proposed dataset. The main contributions of this paper are as follow:
•	This new dataset is the first sizable dataset for handwritten Ethiopic text-image recognition and it can encourage further research on Ethiopic script recognition.
•	The author assessed the human-level performance of multiple participants in HHD-Ethiopic dataset to establish a baseline for comparison with machine learning models.
•	The author evaluate several state-of-the-art OCR methods on the HHD-Ethiopic dataset.

**Strengths:**

1.	This paper collects and analyzes a new historical handwritten Ethiopic dataset, which is benefit to future research.
2.	This paper compare several popular OCR methods on the proposed new dataset. And it tries performing a fair comparison between human and machine performance on historical handwritten Ethiopic scripts recognition task.

**Weaknesses:**

1.	The structure of Table 2 and its description in the bottom of page 6 is not match: According to the Table, the Test-set-I is the IID data, which should be Annot-V with 26.56% CER and 24.56% NED; the Test-set-II is the Annot-VI. And it is better to add a row displaying the average performance.
2.	Because of the error in Section 4.1, the Figure 5 and the conclusion inferred from it are also wrong (i.e. HPopt-Attn-CTC cannot surpass human performance on Test-set-II).
3.	According to the previous two reasons, I think the experiments of the paper are insufficient.
4.	This work did not propose a new approach to recognize with the specific Historical handwritten Ethiopic script.

**Questions:**

Researchers often compare the performance of recognition methods for other languages using the recognition accuracy. Why not include this metric?

---

> ### Author Response · Authors · 2023-11-18
> **Response to reviewer HsR1**
>
> We appreciate your valuable feedback. Thank you.
>
> [**Correction on structure of table 2 and add a row for average performance**]: we have corrected it (the middle horizontal line is moved four steps down, and  inserted one additional row for average performance of each group. Furthermore, we have updated the corresponding values of CER and NED in the text.
>
> [**Mismatch form section 4.1, the Figure 5 and the conclusion inferred from it are also wrong**]: After fixing the table structure in Section 4.2 (Table 2), the remaining inferences drawn from this table are now accurate.
>
> [** According to the previous two reasons, I think the experiments of the paper are insufficient**]: Although we have previously shared the source code, trained models, and dataset for reproducibility, an “\hline” mistake in the LaTeX table code caused confusion in a section of the paper derived from this table. We have now corrected this error and the experiments are corrected.
>
> [**New approach for historical handwritten Ethiopic script**]  as mentioned in the introduction section, there is lack of historical handwritten dataset for Ethiopic, the main goal of this paper is to create the dataset and evaluate it with SOTA OCR models. In addition, we evaluate human-level performance as a benchmark.
>
> [**Why not include accuracy as a metric?**] We are using standard metrics (e.g  e.g https://arxiv.org/pdf/1909.07741.pdf) for sequence to sequence comparisons, where 3 types of errors occur at the character level: insertions, substitutions, and deletions. The 1-CER (CER=Character Error Rate) is as close as it gets to an accuracy (taking into account the 3 types of possible errors). At the sentence level, the accuracy is usually zero because there are usually a few character-level mistakes. Of course, if a language model or lexicon would be integrated in our system to mitigate some of the mistakes, the sentence-level error would go down, but this is no in the scope of this research.

---

### Official Review · Reviewer_idkx · 2023-11-01

**Soundness:** 2 fair
**Presentation:** 3 good
**Contribution:** 3 good
**Rating:** 5
**Confidence:** 4

**Summary:**

This paper presents a new dataset for historical handwritten Ethiopic script. The dataset contains approximately 80k text lines extracted from scanned document from the 18th to the 20th century. The paper describes the challenges of Ethiopic script, and the data collection and annotation process. Baseline recognition results are reported as character error rates, evaluating a number of state-of-the-art techniques in handwriting recognition, as well as the human error rate on the proposed test splits.

**Strengths:**

First, the paper presents a new dataset for an under-resourced script, made of historical documents: this is quite valuable for a more general inclusion of all languages and eventually allow to process more archives.
This seems to be even more valuable given the high error rate of human readers for these documents, even when they are familiar with Ethiopic characters. Systems derived from that dataset could therefore be a good help for archivists.
Finally, the paper provides some baselines with existing handwriting recognition methods.
The paper is globally well written and easy to follow.

**Weaknesses:**

There are maybe sometimes too many details in the text, or repeated statements, which could be shortened in favor of either more analysis of the data, a more detailed description of the challenges of Ethiopic scripts (e.g. some parts of App. A are very interesting and could fit in the main part of the paper).
The fact that some datasets exist for Ethiopic script (described in App. B) should appear in Section 2, where the reader is let to think that no such dataset exist. Moreover, it would be interesting to see how the models presented in the experiment section would perform on these other datasets, or to include the models used in these other papers in the baseline, as they might address the challenges of Ethiopic script. (for example, Abdurhaman et al. report a CER of less than 2% on their dataset)
In a paper proposing a new dataset, I would expect more statistical analysis or the proposal of a method to address the specific challenges of the new dataset. In particular, the human error rate seems quite high, which at the same time makes the dataset interesting but begs the question of the quality of the ground-truth. How would the human performance be with access to reference material? Do the annotator make the same errors or different ones? Do the model make the same mistakes as the human evaluation? For the training set labeling, is there a measure of, for example, inter-annotator agreement?
The paper is interesting and the proposal of a new dataset valuable, but it could be more suited to other venues like DAS, ICFHR, ICDAR or maybe ICML

**Questions:**

Why are CRNN, ASTER, SVTR, etc. models worse than the CTC ones despite being larger? In appendix C we learn that they were not trained to the end due to the lack of resources, but that makes the result confusing, if not misleading if they are understood as baselines.

In the first test set, did you make sure that the same writer or document do not appear in the training and test set?

Minor remarks:
  - the format of citations needs to be fixed
  - the "remark" column in Table I is not necessary
  - Sec. 3.1: "we have generate" -> generated
  - p5. CTC, attention and transformers are put in parallel when they correspond to different things (loss, mechanism, model)

---

> ### Author Response · Authors · 2023-11-18
> **Response to reviewer [idkx]**
>
> Thank you for taking the  time and effort to reviewing our paper.  We address your concerns and questions as follows.
>
> [**Content rearrangement**] we reorganized contents in appendix A, B moving a portion to section 1 and 2 of the main paper (updates are highlighted )
>
> [**Evaluate models proposed in baseline papers on other Ethiopic datasets**] we evaluate our HHD-ethiopic dataset using models proposed for Ethiopic script by Dikubab et al. (2022) and Belay et al. (2019a) and present the result in Table 3. However, neither the source code nor pretrained models are  provided by Abdurahman et al. (2021) and it is difficult to reproduce the results and use the model to evaluate our dataset. In addition, the dataset of Abduraman is a word level dataset extracted from just 1574 lines of modern Amharic handwritten texts, which follow different writing structures (Example in modern ethiopic writing space is a separator while both space and  “two dots” are used as a separator in historical scripts–see Figure 11 in appendix section).  To add more, we evaluated our dataset using a similar CNN-RNN-CTC architecture proposed by Abdurahman et al. (2021), although we haven't accessed the implementation details provided by Abdurahman (See Table 3 in the main paper).
>
> [**Question on human performance and details of the annotation**]. We provided the details of the annotation process (in section 3.1 and appendix B, phase I&II) and human-level performance evaluation (in section 3.2 and appendix B, phase III).  To make it more clear, below is the answer for each question:
>  >**How would the human performance be with access to reference material?** Those individuals participated in the human-level performance have no access for  references. The only individuals having access to the reference are people participating in Ground-truth annotation and their supervisors ( see  section 3.1 and appendix B, phase I).
>
> >**Do the annotators make the same errors or different ones?**  Annotators make different errors in the case of tests, as each text line is annotated by more than one person. However, in the training set, due to the cost associated with annotating each line by many annotators, each text-line image was annotated by an individual.
>
> >**Do the model make the same mistakes as the human evaluation?**  Based on our observations during the experiment and the sample predictions shown in Figure 7, the mistakes differ. Both models and humans made substitution errors, but the model tends to have a higher number of insertions and deletions.
>
> >**For the training set labeling, is there a measure of, for example, inter-annotator agreement?**  For the training set, since each text-line is annotated by an individual annotator we haven’t measured inter-annotator agreement for the training set. However we measure the agreement between the supervisors and the expert for the test set (see the revised version of the main paper, section 4.1 highlighted in yellow).
>
> [**The paper is interesting and the proposal of a new dataset valuable, but it could be more suited to other venues like DAS, ICFHR, ICDAR or maybe ICML**],  Thanks for the suggestion. We  found that the "datasets and benchmarks" theme at ICLR 2024 aligns well with our work, and that's why we're interested in submitting to this venue.
>
> [**Why are CRNN, ASTER, SVTR, etc. models worse than the CTC ones despite being larger? In appendix C we learn that they were not trained to the end due to the lack of resources, but that makes the result confusing, if not misleading if they are understood as baselines**].  As discussed in Appendix C.2, we conducted runs for all models with a limited wall time, less than 24 hours, regardless of their complexity. The initial CTC-based models, with smaller parameters, underwent more epochs, whereas models with larger parameters could not. Furthermore, we implemented the CTC-based model using optimized hyperparameter values through Bayesian optimization (refer to section 3.2 of the main paper), resulting in improved outcomes.
>
> [**In the first test set, did you make sure that the same writer or document does not appear in the training and test set?**].  Text-line image samples in the IID test sets are randomly selected from the distribution using sklearn train-test split. For OOD, only 18th-century documents are chosen without mixing the same documents.
>
> [**Minor remarks**] We corrected and highlighted the changes.

---

### Author Response · Authors · 2023-11-18
**General comment**

We thank the reviewers for their time and effort in reviewing our work. Their recognition of the dataset's significance and the importance of evaluating human-level performance is appreciated. The reviewers raised concerns about details of annotation, human-level performance and citation of related works.

We have addressed these concerns by conducting additional experiments, revising the paper, and including  figures and text. We provided details on annotation and human-level performance. We also ran additional experiments  such as TrOCR and tested both Ethiopic-Eritrea OCR models from Transkribus models and reported their recognition performance. We believe these updates in the main paper, along with the detailed discussions in the appendix, enhance the clarity and validity of our work.

---

### Meta-Review · Area_Chair_AeCV · 2023-12-05

**Metareview:**

The paper proposes a data set of historic, handwritten Ethiopic script.

The paper is not favored by the reviewers, and the discussion has been around the technical details of the paper. The main issue is that the paper does not demonstrate "novelty". Novelty here is not just about being the first, but more about 1) what aspect of Ethiopic script do others want to study but **cannot** be studied due to a lack of resources 2) how this particular data set solves that issue so that others can start studying.

The above problem leads to reviewers questioning whether ICLR is the right venue, what impact this data set would create, and comparison to other data sets.

Overall, ICLR very much welcome a paper that proposes a data set, but there needs to be technical depth in the problem that the data set is solving, not just because there isn't much data.

**Justification For Why Not Higher Score:**

The paper lacks the argument and the demonstration that the data set solves a problem.

**Justification For Why Not Lower Score:**

Rejection is the lowest score.

---

### Decision · Program_Chairs · 2024-01-16

Reject